# Total Fatty Acid Analysis of Human Blood Samples in One Minute by High-Resolution Mass Spectrometry

**DOI:** 10.3390/biom9010007

**Published:** 2018-12-27

**Authors:** Sandra F. Gallego, Martin Hermansson, Gerhard Liebisch, Leanne Hodson, Christer S. Ejsing

**Affiliations:** 1Department of Biochemistry and Molecular Biology, Villum Center for Bioanalytical Sciences, University of Southern Denmark, 5230 Odense, Denmark; sandra.fnn@gmail.com (S.F.G.); martinher@bmb.sdu.dk (M.H.); 2Institute of Clinical Chemistry and Laboratory Medicine, Regensburg University Hospital, 93053 Regensburg, Germany; gerhard.liebisch@klinik.uni-regensburg.de; 3Oxford Centre for Diabetes, Endocrinology and Metabolism, University of Oxford and Oxford NIHR Biomedical Research Centre, Churchill Hospital, Oxford OX3 7LE, UK; leanne.hodson@ocdem.ox.ac.uk; 4Cell Biology and Biophysics Unit, European Molecular Biology Laboratory, 69117 Heidelberg, Germany

**Keywords:** total fatty acid analysis, human blood plasma, shotgun lipidomics, high resolution mass spectrometry, Orbitrap

## Abstract

Total fatty acid analysis is a routine method in many areas, including lipotyping of individuals in personalized medicine, analysis of foodstuffs, and optimization of oil production in biotechnology. This analysis is commonly done by converting fatty acyl (FA) chains of intact lipids into FA methyl esters (FAMEs) and monitoring these by gas-chromatography (GC)-based methods, typically requiring at least 15 min of analysis per sample. Here, we describe a novel method that supports fast, precise and accurate absolute quantification of total FA levels in human plasma and serum samples. The method uses acid-catalyzed transesterification with ^18^O-enriched H_2_O (i.e., H_2_^18^O) to convert FA chains into ^18^O-labeled free fatty acids. The resulting “mass-tagged” FA analytes can be specifically monitored with improved signal-to-background by 1 min of high resolution Fourier transform mass spectrometry (FTMS) on an Orbitrap-based mass spectrometer. By benchmarking to National Institute of Standards and Technology (NIST) certified standard reference materials we show that the performance of our method is comparable, and at times superior, to that of gold-standard GC-based methods. In addition, we demonstrate that the method supports the accurate quantification of FA differences in samples obtained in dietary intervention studies and also affords specific monitoring of ingested stable isotope-labeled fatty acids (^13^C_16_-palmitate) in normoinsulinemic and hyperinsulinemic human subjects. Overall, our novel high-throughput method is generic and suitable for many application areas, spanning basic research to personalized medicine, and is particularly useful for laboratories equipped with high resolution mass spectrometers, but lacking access to GC-based instrumentation.

## 1. Introduction

Total fatty acyl (FA) analysis of human blood samples (i.e., lipotyping) is important for the diagnosis of essential fatty acid deficiency [1] and inborn errors of fatty acid metabolism [2], for lipid-based risk stratification at the population-level, and for monitoring the metabolic health of individuals [3,4,5,6]. Moreover, total FA analysis is also in widespread use in other fields of research, including optimization of oil production by microorganisms and plants [7,8], and monitoring the composition of foods [9]. These application areas typically require a high sample throughput, which inherently benefit from simple, high-throughput routines supporting accurate, precise, and absolute quantification of total FA levels.

Total FA analysis is typically carried out using gas-chromatography (GC)-based methods. The underlying methodology is straightforward and involves using either acid- or base-catalyzed reactions to convert FA chains of intact lipids and non-esterified fatty acids into volatile FA methyl ester (FAME) species that can be detected using GC coupled to a flame ionization detector (GC-FID) [10,11] or a mass spectrometer (GC-MS) [10,11,12,13]. By using different chromatographic parameters (incl. column materials, gradients) and detection principles, it is possible to monitor FA analytes at different levels of structural resolution, which typically scales with increased analysis time and technical diligence of the operator [13,14]. As such, total FA analysis ranges from the identification of the total number of acyl carbon atoms and double bonds (e.g., FA 18:1), requiring about 15 min of analysis per sample, to the separation of distinct FA isomers having different positions and configurations of double bonds (e.g., FA 18:1(9Z) and FA 18:1(11E)), requiring up to 125 min of analysis [13]. Thus, the downside of increasing structural resolution is a significant increase in analysis time. Another potential downside of GC-based approaches is that accurate quantification requires multiple isotope-labeled internal standards (isotope dilution method) as well as generation of multiple calibration curves [13,14]. Notably, in clinical settings, and in other application areas, one needs to strike a unique balance between the analysis time, the depth of structural resolution, and the required sample throughput.

Total FA analysis can also be performed by electrospray ionization-based methods coupled to upfront liquid chromatography (i.e., LC-MS) or by direct infusion (shotgun) MS. Although GC-MS is considered the gold-standard method for total FA analysis, the detection capabilities and analysis time of LC-MS-based methods parallel those of GC-MS. Samples require chemical derivatization, or hydrolysis to free fatty acids, but the methodology can be highly sensitive [15] and support metabolic flux analysis [16]. More recently, a method for total FA analysis using multiplexed isobaric tagging and 40 min of LC-MS was described [17]. Although the structural resolution provided by this method is no match to that of GC-MS (only identification of total number of carbon atoms and double bonds), the potential multiplexing of up to ten samples drives this approach towards high-throughput.

FA analysis by direct infusion (shotgun) MS is also relatively simple and fast, and, in comparison to GC- and LC-based approaches, devoid of any sample carry-over when using automated chip-based sprayers for injecting individual samples [18]. Various derivatization methods for analyzing non-esterified fatty acids by direct infusion MS have been developed [19,20]. These methods, however, are not directly applicable for monitoring the composition of esterified FA analytes derived from intact lipid molecules. Nevertheless, a particular virtue of these methods is that they enable MS/MS-based structural characterization of double bond positions. We have recently used high resolution Fourier transform mass spectrometry (FTMS) in negative ion mode for the detection of non-esterified fatty acids as carboxylate anions and quantification of these in a larger cohort of plasma samples [21]. This work suggested that negative ion mode FTMS analysis could also be used for total FA analysis, and potentially be faster and more sensitive than most high throughput-based GC and LC methods.

Here, we describe a novel method for the quantification of total FA levels in human plasma and serum samples. The method uses acid-catalyzed hydrolysis of intact lipids in the presence of ^18^O-enriched water (i.e., H_2_^18^O). This reaction yields ^18^O-labeled free fatty acid species that can be specifically detected and quantified by automated high resolution FTMS analysis on an Orbitrap-based instrument coupled to a chip-based nanoelectrospray ionization source. Notably, the reaction can be carried out directly with diluted plasma and serum without prior lipid extraction, thereby bypassing potential biases deriving from sample extraction. We show that (i) negative ion mode analysis of deprotonated FA analytes is exceptionally sensitive, (ii) quantitative analysis can be performed in only 1 min, (iii) the method is accurate and precise, and (iv) that high resolution FTMS analysis with a resolving power above 400,000 (full width at half maximum, FWHM) affords baseline separation of ^13^C, ^2^H, and ^18^O isotopologues, which offers a new analytical avenue for metabolic flux analysis. This method is generic and will be useful for laboratories equipped with high resolution mass spectrometers, but lacking access to GC-based instrumentation, and for quantifying total FA levels in a variety of biological sample matrices.

## 2. Materials and Methods

### 2.1. Chemicals and Standards

Acetonitrile, *n*-hexane, methanol, 2-propanol, and water were purchased from Biosolve BV (Valkenswaard, The Netherlands). Chloroform was from Rathburn Chemicals (Walkerburn, UK). Ammonium formate was from Sigma-Aldrich (Buchs, Switzerland). All solvents and chemicals were HPLC grade. Sulfuric acid and butylated hydroxytoluene (BHT) were from Sigma-Aldrich (Steinheim, Germany), ^18^O-labeled water (i.e., H_2_^18^O) was from Medical Isotopes (Pelham, NH, USA). Lipid standards PE 15:0/18:1(+^2^H_7_) (1-pentadecanoyl-2-oleoyl(d7)-sn-glycero-3-phosphoethanolamine) and LPC 16:0(+^2^H_3_) (1-palmitoyl-d3-sn-glycero-3-phosphocholine) were obtained from Avanti Polar Lipids (Alabaster, AL, USA). A FAME standard mixture, with equal amounts of FAME 16:0, 18:0, 16:1, 18:1, 18:2, 18:3n−3, 18n−6, 20:3, 20:4, 20:5, 22:4, 22:5n−3, 22:5n−6 and 22:6, was obtained from Larodan Fine Chemicals (Solna, Sweden).

### 2.2. Human Plasma Samples

National Institute of Standards and Technology (NIST) Standard Reference Material (SRM) 1950 and 2378 were used for validation. In short, SRM 1950 was collected from 100 fasted individuals in the age range of 40–50 years and represents the plasma average composition of the US population as defined by race, sex, and health. SRM 2378 consists of 3 serum materials collected from: (1) donors who did not take either fish or flaxseed oil supplements for one month prior to collection, (2) donors who took flaxseed oil supplements for a minimum of one month prior to collection, and (3) donors who took fish oil supplements for a minimum of one month prior to collection.

Plasma samples from five normoinsulinemic and five hyperinsulinemic subjects were obtained as previously described [22]. In short, after an overnight fast the subjects had a cannula inserted into an antecubital vein, and baseline (0 h) blood plasma samples were taken. Subjects were then fed a mixed meal containing 200 mg of ^13^C_16_-labeled palmitic acid (FA 16:0(+^13^C_16_)). Another blood plasma sample was taken after 6 h, shown previously to yield the highest incorporation of FA 16:0(+^13^C_16_) into plasma triacylglycerols [22]. The study was approved by Portsmouth Clinical Research Ethics Committee (REC 12/SC/0267), and all subjects gave written informed consent. The study was performed in accordance with the Helsinki Declaration.

### 2.3. Sample Preparation

Plasma and serum, 15 µL, were mixed with 210 µL of 155 mM ammonium formate. Fifteen µL of this mixture (equivalent to 1 µL of undiluted blood sample) was transferred into a 1.1 mL glass vial (La-Pha-Pack GmbH) with 7 µL of 162.3 µM PE 15:0/18:1(+^2^H_7_) added as internal standard, and vacuum evaporated. Next, the sample was dissolved in 80 µL of freshly prepared 0.75 M H_2_SO_4_ in acetonitrile/H_2_^18^O (9:1, v/v) containing 50 µg/mL BHT. For hydrolysis, the sample was placed in a ThermoMixer (Eppendorf, Wesseling-Berzdorf, Germany) for 5 h at 90 °C and 750 rpm. After cooling, fatty acids were extracted with 180 µL of *n*-hexane by mixing for 10 min at 1400 rpm and centrifugation for 5 min at 1000 *g*. The fatty acid extraction with *n*-hexane was repeated, and the combined extracts were vacuum evaporated.

### 2.4. Mass Spectrometric Analysis

The total FA extracts were dissolved in 500 µL chloroform/methanol/2-propanol (1:2:4, v/v/v) containing 0.75 mM ammonium formate and loaded in a 96-well plate (Eppendorf, Hamburg, Germany). Ten µL of each fatty acid extract was infused using the robotic nanoflow ion source TriVersa NanoMate (Advion Biosciences, Ithaca, NY, USA) and analyzed in negative ion mode using an Orbitrap Fusion Tribrid mass spectrometer (Thermo Fisher Scientific, San Jose, CA, USA). Ionization voltage was −0.96 kV and back pressure was 1.25 psi. The temperature of the ion transfer tube was 275 °C. S-lens radio frequency level was set to 60%. Negative ion mode FTMS analysis was performed in m/z range 150–420. Each sample was analyzed for 1 min. All full scan FTMS data were acquired in profile mode, using a max injection time of 100 ms, automated gain control for an ion target of 10^5^, three microscans (to reduce file size), and a target resolution setting of 500,000.

### 2.5. Lipid Identification and Quantification

Carboxylate anions of ^18^O-labeled and unlabeled (^16^O) FA analytes detected by FTMS analysis with a mass accuracy better than ±2.5 ppm were identified and quantified using ALEX^123^ software [23]. FA species were quantified by summing the intensities of doubly (^18^O_2_) and singly (^16^O^18^O) tagged FA analytes, normalizing to the sum intensity of doubly (^18^O_2_) and singly (^16^O^18^O) tagged internal standard FA 18:1(+^2^H_7_), and multiplying by the spike amount of PE 15:0/18:1(+^2^H_7_). To minimize bias from FA-contaminants present in reagents and other materials (see below), a blank correction was performed by subtracting the average amount of FA detected in reaction blanks from the amounts quantified in the samples. Statistical analysis, paired and unpaired *t*-tests, was performed using SAS 9.3 (SAS Institute, Cary, NC, USA).

### 2.6. FAME Analysis

Human plasma and a reaction blank were analyzed as previously described [24]. Briefly, samples were derivatized using methanolic acetylchloride in *n*-hexane, and FAMEs were extracted using potassium carbonate in water. The total FAME extracts were analyzed by GC-MS using a total run time of 15 min.

## 3. Results and Discussion

### 3.1. Total Fatty Acyl Analysis by Direct Infusion FTMS Analysis

The aim of the current study was to develop a method that supports fast, precise, and accurate absolute quantification of total FA levels in human blood plasma and serum samples using direct infusion-high resolution FTMS analysis. To this end, we adapted the acid-catalyzed transesterification-based protocol commonly used for total FA analysis by GC-based approaches to generate free fatty acids instead of FAMEs. This was achieved by substituting methanol with H_2_O (see Materials and Methods for details). The rationale for this design was that negative ion mode analysis of deprotonated fatty acids (i.e., [M-H]^−^) is highly sensitive, and that quantitative analysis of intact lipids by direct infusion FTMS can be performed using only a few minutes of analysis per sample [25,26,27]. Through meticulous method optimization we devised a method that uses only 1 µL of human plasma or serum, instead of 10–50 µL plasma as required by gold-standard GC-MS-based routines [13,24]. For total FA analysis, the 1 µL of plasma or serum is spiked with a defined amount of internal standard (e.g., PE 15:0/18:1(+^2^H_7_)), followed by evaporating the sample to dryness, hydrolyzing intact lipids using sulfuric acid (H_2_SO_4_) in acetonitrile/H_2_O (9:1, v/v), extracting the free fatty acids with hexane, and analyzing (2% of) the final total FA extract by 1 min of automated direct infusion-high resolution FTMS analysis. Using this approach, in conjunction with hydrolysis in regular ^16^O-rich H_2_O, yields high resolution FTMS spectra with detection of endogenous FAs and internal standard-derived FAs as deprotonated carboxylate ions (Figure 1A). We note that this approach (i.e., full scan FTMS analysis) only allows annotating detected FAs at the “lipid species-level”, indicating the total number C atoms and double bonds in a particular FA analyte.

Having established that our strategy in principle enables fast and simple monitoring of total FA levels, we proceeded to evaluate its performance in terms of analytical accuracy. In our initial assessments we found that especially FA 16:0 and FA 18:0 give rise to high background intensity levels (Appendix A). This analytical caveat prompts poor signal-to-background values for the detection of endogenous FA 16:0 and FA 18:0, which in turn hampers their accurate and precise quantification. Importantly, this caveat is common to all total FA analysis methods, including GC-MS-based routines (Appendix A), and is caused by FA contaminants present even in high-grade organic solvents, glassware and pipette tips used for sample preparation [28]. Notably, we found that the signal-to-background for FTMS analysis of FA 16:0 and FA 18:0 in 1 µL of human plasma was 9.9 and 4.3, respectively (Appendix A). In comparison, the signal-to-background for GC-MS-based analysis of FAME 16:0 and FAME 18:0 in 1 µL of human plasma is approximately 3.3 and 1.8, respectively (Appendix A). Hence, the performance our novel method for monitoring of FA 16:0 and FA 18:0, based on lipid hydrolysis with regular ^16^O-rich H_2_O and FTMS analysis, is comparable to that of GC-MS-based routines.

### 3.2. Hydrolysis with H_2_^18^O Improves Signal-to-Background

To minimize, or potentially remove, the high background intensity from FA 16:0 and FA 18:0 we next explored the efficacy of hydrolyzing intact lipids in the presence of ^18^O-enriched H_2_O (i.e., H_2_^18^O). We deemed that this procedure should reduce the contribution of background by incorporating a “mass tag” label into endogenous FAs with either one or two ^18^O atoms, corresponding to inserting a specific mass offset of 2.0042 and 4.0085, respectively, compared to underivatized, background FA species having two ^16^O atoms. Using this strategy, we first found that the unique mass signature of the ^18^O atom, and using a resolving power in the order of 400,000 (FWHM at m/z 300) for FTMS analysis, afforded specific detection of FA species labeled with either one or two ^18^O atoms with baseline separation from isobaric FA molecule having any configuration of ^13^C, ^12^C, ^2^H, ^1^H and ^16^O atoms (Figure 1B–F). Notably, baseline separation of isobaric FA analytes can be achieved using instrumentation with a mass resolution higher than ~95,000 (FWHM at m/z 300). Assessing the FA 16:0 and FA 18:0 background levels after hydrolysis with H_2_^18^O showed more than a 2.3-fold improvement in the signal-to-background for detection of endogenous FA 16:0 and FA 18:0, respectively (Appendix A). As a safeguard for using H_2_^18^O we also investigated, over a period of 56 days (eight weeks), whether the incorporated ^18^O atoms would scramble during sample storage at −20 °C. This assessment demonstrated that ^18^O-labeled FA analytes are highly stable and do not incorporate ^16^O atoms upon storage (Appendix A). Based on these observations, we concluded that acid-catalyzed lipid hydrolysis with H_2_^18^O and high resolution FTMS analysis is a feasible and specific strategy for monitoring total FA levels in human plasma samples.

### 3.3. Instrument Response Is Independent of Fatty Acyl Chain Length and Double Bond Number

The combined efficiency of ionization and detection of lipid molecules is an important parameter for accurate and absolute quantification. This efficiency is dependent on numerous parameters, including the chemistry of the lipid class, the FA chain length and number of double bonds, the ionization mechanism (i.e., ion source), the type of mass spectrometer and the acquisition method [11,14].

To examine the instrument response for deprotonated FA analytes, we hydrolyzed and analyzed a FAME standard mixture with 14 different species in equal amounts (w/v) (where 18:3 and 22:5 are present as two isomers). This evaluation showed that all deprotonated FA analytes produced an identical instrument response (Figure 2). Importantly, this demonstrates that the instrument response for detection of deprotonated FA species using nanoelectrospray ionization and FTMS analysis is practically independent of FA chain length and the number of double bonds, at least for FAs with chain lengths from 16 to 22. Moreover, this also justifies using the spike of only a single internal standard (e.g., FA 18:1(+^2^H_7_) derived from PE 15:0/18:1(+^2^H_7_)) to quantify the wide range of FA analytes present in human plasma and serum samples.

### 3.4. Dynamic Quantification Range of Total Fatty Acyl Analysis by FTMS

Next, we evaluated the dynamic quantification range of our method. To this end, we prepared a dilution series where the synthetic standard PE 15:0/18:1(+^2^H_7_) was titrated relative to a constant amount of the synthetic standard LPC 16:0(+^2^H_3_). This dilution series was spiked into human plasma samples, which were subjected to direct acid-catalyzed hydrolysis using H_2_^18^O. The resulting total FA extracts were analyzed by 1 min FTMS analysis. To evaluate the dynamic quantification range, we plotted the intensity ratio of FA 18:1(+^2^H_7_) and FA 16:0(+^2^H_3_) as a function of their molar ratio (Figure 3). The response was linear, with a slope value of approximately one across four orders of magnitude and having a detection limit of approximately 0.9 nM in human plasma (corresponding to the molar ratio value of 0.0013 in Figure 3). Thus, our method is approximately two orders of magnitude more sensitive than typical GC-MS-based methods [11]. Based on this result, we conclude that our novel method affords rapid and highly sensitive quantification of FAs in human plasma. We note that the instrument response for FA 15:0 was also linear, with a slope value of approximately one, but only across three orders of magnitude since human plasma contains low amounts of this odd-chain FA (Appendix A).

### 3.5. Accurate Fatty Acyl Quantification by High Resolution FTMS Analysis

To assess the accuracy of our method, we analyzed the NIST human plasma standard reference material (SRM) 1950. This reference material has previously been used for compiling consensus values on the absolute levels of FAs in human plasma and also for comparing the performance of several laboratories using GC- and LC-based approaches for total FA analysis [29]. To benchmark the performance of our method, we subjected the SRM 1950 plasma to acid-catalyzed hydrolysis with H_2_^18^O and analyzed the total FA extracts by direct infusion FTMS analysis. Overall, we found that estimates of total FA levels by our method were in good agreement with the SRM 1950 consensus values, and also that the accuracy of our method was in several instances superior to that of certain laboratories using GC- or LC-based methods for analysis (Figure 4). In the comparison to consensus values, we found that the most abundant FA 18:2 had a subtle difference of −7%. For the most abundant monounsaturated FA 16:1 and FA 18:1 the accuracy of quantification differed by 8% and 13%, respectively. For polyunsaturated FAs, such as FA 20:4, 22:5 and 22:6, the accuracy of quantification differed by −7%, 2% and −22%, respectively. For FA 16:0 and FA 18:0, having the highest background intensity (see above), we found that the estimated concentration differed by −37% and −13%. We note that this discrepancy is related to the difference in signal-to-background mentioned above (Appendix A) and the higher accuracy of estimating the FA background level by our approach. As such, our method seems to provide a more accurate estimate of the relatively high FA 16:0 background level than GC-based routines, and using this higher background amount to estimate the endogenous FA 16:0 level yields, by background subtraction, a lower endogenous FA 16:0 concentration than listed in the SRM 1950 Certificate of Analysis. Overall, based on our detailed assessment of the quantification accuracy, we conclude that our novel method affords accurate absolute quantification of total FA levels in human plasma, and its performance is comparable, or at times superior, to that of numerous laboratories using gold-standard GC- and LC-based routines for total FA analysis.

### 3.6. Total Fatty Acyl Analysis by FTMS Is Precise and Applicable for Routine Analysis

Next, we determined the analytical precision of our novel method for total FA quantification in human plasma samples. To this end, we used two types of plasma samples: i) a pooled plasma sample from fasted normoinsulinemic subjects having a total FA concentration of 11.5 mM (normal lipid level), and ii) a pooled plasma sample from 6 h postprandial hyperinsulinemic subjects having a total FA concentration of 24.8 mM (high lipid level). To determine the intra-day and inter-day precision, we independently prepared and analyzed five total FA extracts of these two types of plasma samples. This sample preparation and analysis was repeated on three independent days. The precision on individual days (intra-day precision) showed a coefficient of variation (CV) always below 8.6% and 11.3% for the normal and high lipid level samples, respectively (Table 1). The inter-day (day-to-day) precision showed CV values always below 11.1%. These performance characteristics demonstrate that our novel method for total FA analysis, based on acid-catalyzed hydrolysis with H_2_^18^O and direct FTMS analysis, is similar to that of gold-standard approaches using FAME derivatization and GC-based analysis [11], and also to that of routine methods used for quantification of other clinically-relevant lipid molecules [25,30,31].

### 3.7. Lipotyping of Human Serum Samples by Total Fatty Acyl Analysis

Next, we evaluated the efficacy of our method for determining differences in total FA levels in samples from humans subjected to dietary interventions. For this purpose, we used the NIST SRM 2378 [32], a reference material of pooled human serum from three groups of healthy individuals: (i) subjects supplemented with fish oil (enriched in eicosapentaenoic acid, FA 20:5, and docosahexaenoic acid, FA 22:6), (ii) subjects supplemented with flaxseed oil (enriched in linoleic acid, FA 18:2), (iii) and control subjects (no dietary intervention). Total FA analysis of these serum samples demonstrated again that our method is accurate and capable of reproducing expected total FA levels listed in the Certificate of Analysis (Appendix A). Here the main differences were found for FA 16:0 and FA 20:4, which were in averaged under-estimated by 40% and over-estimated by 42%, respectively. Furthermore, our results also showed that serum from subjects supplemented with fish oil had increased levels of polyunsaturated FAs, including up to 5-fold and 2-fold higher FA 20:5 and FA 22:6 concentrations, respectively, compared to control serum (Figure 5). In addition, we also observed a 1.4-fold higher concentration of FA 18:1 in individuals supplemented with flaxseed oil. Taken together, these results consolidate that our novel total FA analysis method supports accurate monitoring of differences in FA composition in fasted human blood samples and is applicable for personalized lipotyping similar to GC- and LC-based approaches.

### 3.8. Quantitative Analysis of Stable Isotope-Labeled Fatty Acyls in Human Plasma

Based on the analytical capability of our high resolution Orbitrap-based mass spectrometer to baseline-separate individual FAs having distinct composition of isotopologues (e.g., ^12^C, ^13^C, ^1^H, ^2^H, ^16^O and ^18^O atoms), we next decided to explore whether our method would also be applicable for metabolic studies in humans. Such studies are commonly done by feeding human subjects with stable isotope-labeled fatty acids, sampling blood plasma at defined time points after intake, and performing time series analysis of tracer incorporation using GC-MS [22,33]. To evaluate the performance of our platform for such studies, we analyzed plasma samples from ten individuals, classified as normoinsulinemic (*n* = 5) and hyperinsulinemic (*n* = 5). The plasma samples were collected at baseline (time point 0 h) and 6 h after the individuals had consumed a mixed meal containing 200 mg of ^13^C_16_-labeled palmitic acid (i.e., FA 16:0(+^13^C_16_)) [22]. Notably, previous analysis has shown that after 6 h the plasma pool of FA 16:0(+^13^C_16_) is primarily present as a non-esterified fatty acid (~1 µM) and incorporated into triacylglycerol (~2 µM) [22].

Total FA analysis using acid-catalyzed H_2_^18^O hydrolysis and 1 min of FTMS analysis showed that deprotonated FA 16:0(+^13^C_16_) labeled with one or two ^18^O atoms were detected in all plasma samples collected at the 6 h time point and not at baseline (Figure 6). This shows that our method is specific and effectively free of noise for monitoring the uptake of FA 16:0(+^13^C_16_) into blood plasma. We also searched the mass spectral data for whether any metabolic products of FA 16:0(+^13^C_16_) could be detected, including FA 16:1(+^13^C_16_), FA 18:0(+^13^C_16_) and FA 18:1(+^13^C_16_). However, none of these metabolites could be detected, potentially due to the limit of detection of our method. The total FA 16:0(+^13^C_16_) concentration in plasma from the normoinsulinemic and hyperinsulinemic subjects was estimated to be 4.9 ± 2.4 µM and 5.4 ± 3.1 µM, respectively. Statistical analysis showed no significant difference in the total FA 16:0(+^13^C_16_) concentration (unpaired, two-sided *t*-test, *p*-value = 0.7). Importantly, this result corroborates the previous findings by Pramfalk et al. (2016) reporting a similar total concentration of FA 16:0(+^13^C_16_) in the plasma samples and no difference between the normoinsulinemic and hyperinsulinemic subjects at the 6 h time point. However, the previous study did find a significant difference in the temporal profile of non-esterified FA 16:0(+^13^C_16_) level between the two groups of subjects.

Next, we investigated whether the plasma concentration of other (unlabeled) FA species showed any major differences across the two time points and between the two groups of subjects. Statistical analysis of time-dependent changes in absolute FA concentrations within each group showed that only three FAs changed significantly; FA 14:0 (in both groups), FA 18:1 (only in normoinsulinemic subjects) and FA 16:0(+^13^C_16_) (in both groups) (paired, two-sided *t*-test, *p*-value < 0.05). No significant differences were found between the two groups (unpaired, two-sided *t*-test, *p*-value < 0.05). Notably, this relatively low frequency of significant differences correlates to a relatively high variance in total FA concentration among all the samples (Appendix A).

As an alternative approach to pinpoint differences, we transformed the concentration data to relative mol%, thereby reducing the influence of variations in the total FA concentration. Repeating the statistical analysis revealed that the relative proportions of 18 out of 31 monitored FA species change significantly in either of the two groups (Figure 7) (paired, two-sided *t*-test, *p*-value < 0.05). The three common and most significantly changing FAs were FA 16:1, FA 14:0 and FA 18:3, all of which were reduced 6 h after food intake. The analysis also pinpointed a specific subset of polyunsaturated FAs that are significantly changing only in the normoinsulinemic subjects (i.e., FA 22:4, FA 20:3, FA 22:5, FA 18:2, and FA 16:2), and four distinct FA species, primarily saturated and monounsaturated very long chain FA species, FA 24:1, FA 24:0, FA 22:0, and FA 12:0, that are significantly changing only in the hyperinsulinemic subjects. Finally, the statistical analysis of differences between the normoinsulinemic and hyperinsulinemic subjects revealed that only the relative change of FA 16:1 is statistically significant (unpaired, two-sided *t*-test, *p*-value = 0.04). Here, the temporal reduction in the FA 16:1 level is 1.5-fold higher in normoinsulinemic as compared to hyperinsulinemic subjects. Taken together, these results highlight that our new method lends itself to also lipid metabolic studies in humans, and other mammals, with the ability to reproducibly and specifically monitor incorporation of stable isotope-labeled FA tracers as well as changes in steady-state FA levels following food intake.

## 4. Conclusions

Here, we described the development and validation of a novel method that supports fast, precise, and accurate absolute quantification of total FA levels in human plasma and serum samples. The method uses acid-catalyzed hydrolysis of intact lipids in the presence of ^18^O-labeled H_2_O (i.e., H_2_^18^O) and therethrough the liberation of FA chains as ^18^O-labeled analytes. These “mass-tagged” FA analytes can be specifically detected, with improved signal-to-background, using automated chip-based nanoelectrospray ionization and only 1 min of high resolution FTMS analysis on an Orbitrap Fusion mass spectrometer. We demonstrate that our novel method has similar, or improved, analytical performance in terms of accuracy and precision compared to numerous laboratories using gold-standard GC-based methods for total FA analysis (Figure 4). We deem that our novel method should prove highly useful for laboratories equipped with high resolution mass spectrometers, but lacking access to GC-based instrumentation.

We also demonstrate that our method is applicable for specific tracking of stable isotope-labeled FA tracers in humans. In particular, our method demonstrated that the total concentration of ^13^C_16_-labled FA 16:0 in plasma 6 h after intake was similar in normoinsulinemic and hyperinsulinemic subjects. In contrast, the analysis of steady-state FA levels showed that a subset of polyunsaturated FAs, including FA 18:2, FA 20:3, FA 22:4 and FA 22:5, were significantly reduced only in normoinsulinemic subjects and not changed in hyperinsulinemic subjects. Conversely, we found a subset of very long chain FA species, including FA 24:1, FA 24:0 and FA 22:0, that were significantly reduced only in the hyperinsulinemic subjects. These changes corroborate the notions that polyunsaturated FAs have a positive impact on metabolic health [34] and that very long chain FAs, especially when incorporated into sphingolipids, are associated with increased cardiovascular risk [35,36]. We note, however, that these results have only been obtained by analysis of a relatively low number of subjects (*n* = 5), and that analysis of a larger cohort of subjects, and with more time points, should be performed in order to increase the statistical power of our results. It is tempting to propose that such studies could in future be performed using not only a single FA tracer, but instead a mixture of stable isotope-labeled FA tracers, e.g., FA 18:2n−6, FA 18:3n−3 and FA 16:0, which will simultaneously provide insights into the metabolic trajectories of a wider range of FA species under normoinsulinemic and hyperinsulinemic conditions.

We note that our method does not provide the same level of structural resolution as some of the more time-consuming GC-MS-based approaches. However, it does strike a unique balance between analytical speed and FA molecule coverage. As such, gold-standard GC-MS-based approaches are able to monitor around 44 FAME species using an analysis time of 15 min, i.e., effectively detecting 2.9 FAMEs per minute of analysis [24]. In comparison, our novel approach supports monitoring of at least 31 FA species by 1 min of analysis. We note that, if needed, this analysis time can be reduced to 0.5 min per sample without jeopardizing the FA coverage and the analytical performance. Furthermore, our method can also be adapted, through increased analysis time, to record MS^2^ scans with diagnostic radical FA fragment ions that identify double bond-positional isomers, at least for monounsaturated FA species (e.g., FA 18:1(9) and FA 18:1(11)) [37]. Further developments of this orthogonal approach, for example by combining specialized gas-phase dissociation mechanisms such as OzID [38], should enable the specific detection and quantification of many more FA double bond positional isomers. It is tempting to propose that such technology, if successfully implemented, could potentially be the beginning of making GC-based platforms for routine analysis of total FA levels obsolete.

Finally, our novel method is not only restricted to total FA analysis of human blood samples. We note that our generic, high-throughput method can also be used in many other fields of research. These include routine analysis for assessing food quality, biotechnological application aimed at optimizing oil production in plants and microorganisms, and for biochemical assays of lipid enzymatic activities (e.g., fatty acid synthases, desaturases and elongases). We also note that the use of H_2_^18^O can be replaced with regular ^16^O-rich H_2_O to reduce costs, although this will lead to less accurate estimates of total FA 16:0 and 18:0 levels (1 mL H_2_^18^O for 125 samples costs approx. 88 Euro at the present). Finally, we note that our novel method can also serve as a quality control routine to accurately quantify and verify the concentrations of commercially available lipid standards used for absolute quantification in global lipidomics [21,26,35,39,40,41], instead of using cumbersome colorimetric assays that require high amounts of material to quantify, for example, glycerophospholipids.

## Figures and Tables

**Figure 1 biomolecules-09-00007-f001:**
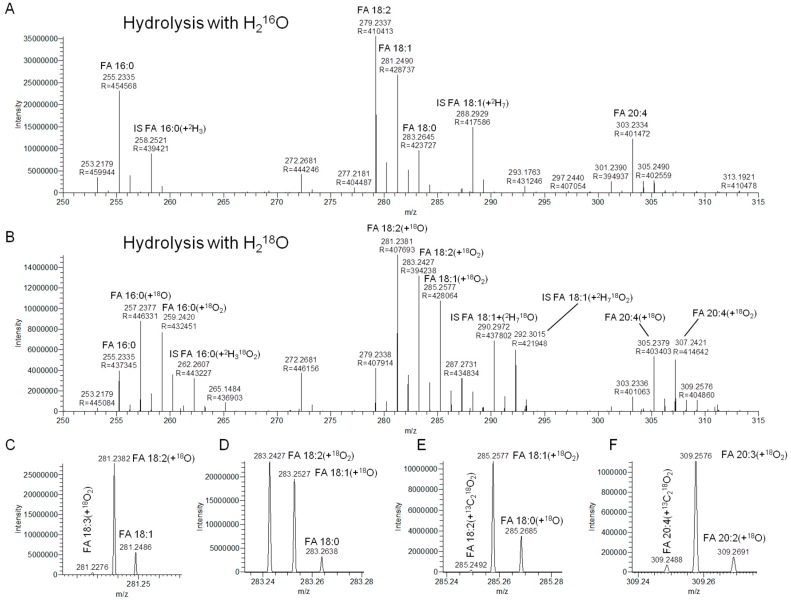
Detection of deprotonated fatty acyl (FA) analytes by 1 min of high resolution Fourier transform mass spectrometry (FTMS). (**A**) Negative ion mode FTMS spectrum of a total FA extract of human plasma using regular ^16^O-rich H_2_O for acid-catalyzed hydrolysis. (**B**) Negative ion mode FTMS spectrum of a total FA extract of human plasma using ^18^O-enriched H_2_O for acid-catalyzed hydrolysis. (**C**–**F**) Selected m/z intervals of the FTMS spectrum shown in B. These m/z intervals show baseline separation of deprotonated FA species with different numbers of double bonds, ^18^O, ^16^O, ^13^C, ^12^C, ^2^H and ^1^H atoms.

**Figure 2 biomolecules-09-00007-f002:**
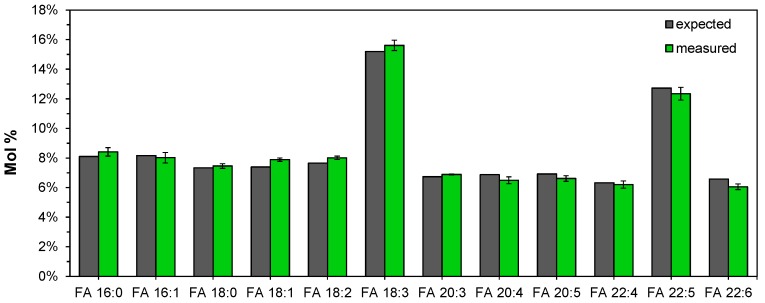
Instrument response for detection of deprotonated fatty acyl (FA) analytes is independent of chain length and number of double bonds. A standard mixture with 14 different NEFA species present in equal amounts (where FA 18:3 and FA 22:5 are present as two isomers, 18:3n−3 and 18:3n−6, and 22:5n−3 and 22:5n−6, respectively), was subjected to acid-catalyzed hydrolysis using H_2_^18^O and Fourier transform mass spectrometry (FTMS). Data represent average ± SD (*n* = 3).

**Figure 3 biomolecules-09-00007-f003:**
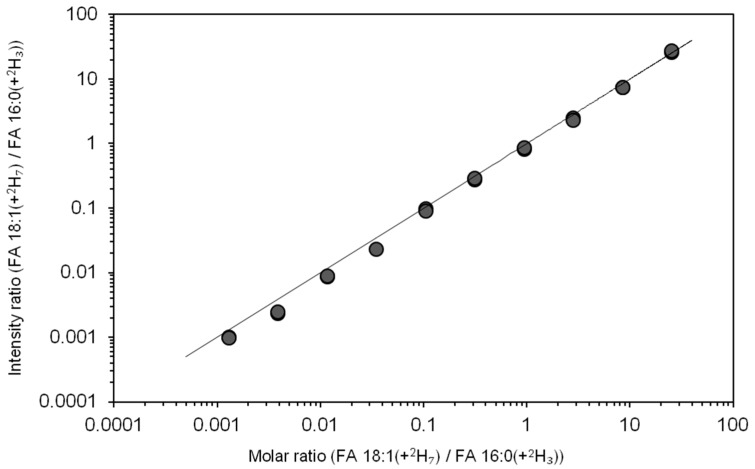
Dynamic quantification range of total fatty acyl (FA) analysis. PE 15:0/18:1(+^2^H_7_) was titrated relative to a constant amount of LPC 16:0(+^2^H_3_), spiked into 1 µL human plasma and subjected to acid-catalyzed hydrolysis using H_2_^18^O. The total FA extracts were analyzed by direct infusion Fourier transform mass spectrometry (FTMS). The x-axis shows the concentration of FA 18:1(+^2^H_7_) relative to the concentration of FA 16:0(+^2^H_3_) (i.e., molar ratio). The y-axis shows the intensity of deprotonated FA 18:1(+^2^H_7_) relative to the intensity of FA 16:0(+^2^H_3_). Depicted values derive from two replicate analyses. The line indicates the linear function with slope 1.

**Figure 4 biomolecules-09-00007-f004:**
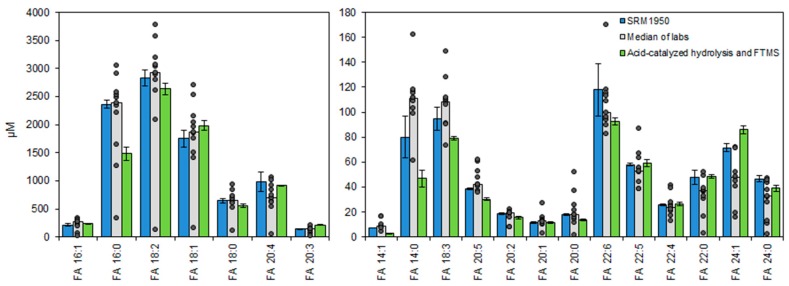
Total fatty acyl (FA) levels in human plasma standard reference material (SRM) 1950. Blue bars represent certified and reference values ± expanded uncertainties (*U*_95%_) as listed in the certificate of analysis for SRM 1950. Grey bars represent medians of average values measured by different laboratories. The concentrations measured by the different laboratories are shown as individual data points. Green bars represent the average concentration ± SD (*n* = 4) measured by Fourier transform mass spectrometry (FTMS) of plasma samples hydrolyzed using sulfuric acid in acetonitrile/H_2_^18^O.

**Figure 5 biomolecules-09-00007-f005:**
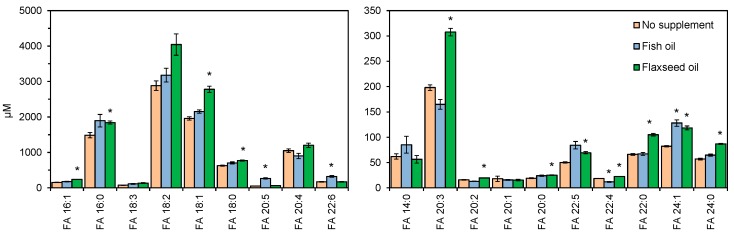
Total fatty acyl (FA) levels in serum from human subjects supplemented with fish oil or flaxseed oil, and subjects receiving no dietary intervention (SRM 2378). Serum lipids were subjected to acid-catalyzed hydrolyzed with H_2_^18^O and analyzed by high resolution Fourier transform mass spectrometry (FTMS). Data represent average ± SD (*n* = 4, independently prepared and analyzed samples). * *p* < 0.001 (unpaired, two-sided Student’s *t*-test with comparison to “No supplement”).

**Figure 6 biomolecules-09-00007-f006:**
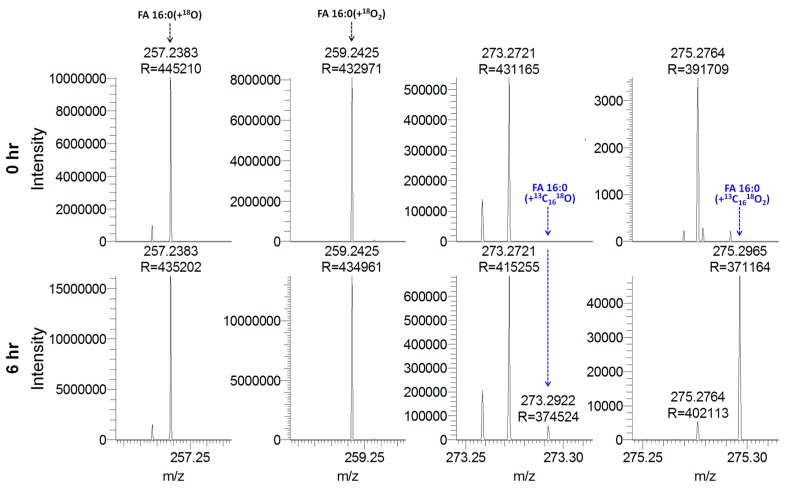
Specific monitoring of fatty acyl (FA) 16:0(+^13^C_16_) in human plasma by high resolution Fourier transform mass spectrometry (FTMS). Representative FTMS spectra of plasma from a normoinsulinemic subject at 0 h (baseline) and 6 h after intake of FA 16:0(+^13^C_16_). Plasma samples were subjected to acid-catalyzed hydrolysis with H_2_^18^O and analyzed by high resolution FTMS analysis. Indicated FA analytes are detected as deprotonated carboxylate anions.

**Figure 7 biomolecules-09-00007-f007:**
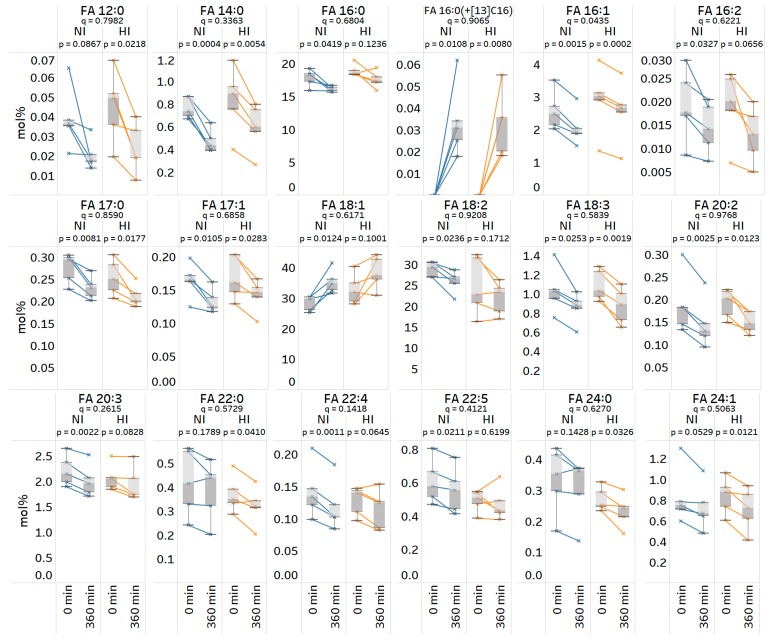
Total fatty acyl (FA) profiles of normoinsulinemic (NI) and hyperinsulinemic (HI) subjects at 0 h (baseline) and 6 h after intake of FA 16:0(+^13^C_16_). Plasma lipids were subjected to acid-catalyzed hydrolysis with H_2_^18^O and analyzed by high resolution Fourier transform mass spectrometry (FTMS). Interconnected data points in box plots represent repeated measurements from five individuals within each group. *p*-values are based on paired, two-sided *t*-tests of differences within each subject group. *q*-values are based on unpaired, two-sided *t*-tests of time-dependent differences between the two groups. *p*- and *q*-values below 0.05 are considered statistically significant.

**Table 1 biomolecules-09-00007-t001:** Intra-day and inter-day precision of total FA analysis in human plasma samples.

		Intra-Day						Inter-Day	
		Day 1 (*n* = 5)		Day 2 (*n* = 5)		Day 3 (*n* = 5)		(*n*= 3 × 5)	
Analyte	Insulin	µM ± SD	CV (%)	µM ± SD	CV (%)	µM ± SD	CV (%)	µM ± SD	CV (%)
FA 16:0	Normal	1852 ± 103	5.5	2085 ± 179	8.6	2028 ± 105	5.2	1988 ± 161	8.1
	High	4239 ± 294	6.9	3798 ± 288	7.6	4373 ± 218	5.0	4137 ± 356	8.6
FA 18:1	Normal	2898 ± 54	1.9	3132 ± 156	5.0	3130 ± 183	5.8	3053 ± 174	5.7
	High	10859 ± 534	4.9	9618 ± 279	2.9	11114 ± 508	4.6	10531 ± 797	7.6
FA 18:2	Normal	3404 ± 133	3.9	3734 ± 192	5.1	3765 ± 162	4.3	3634 ± 227	6.2
	High	3442 ± 275	8.0	* 3031 ± 102	3.4	3640 ± 412	11.3	3396 ± 378	11.1
FA 20:4	Normal	920 ± 24	2.6	948 ± 26	2.8	998 ± 80	8.0	955 ± 58	6.0
	High	1728 ± 95	5.5	1504 ± 56	3.7	1750 ± 111	6.4	1661 ± 142	8.6
FA 20:5	Normal	108 ± 3.6	3.3	115 ± 3.1	2.7	121 ± 10	8.6	115 ± 8.2	7.1
	High	188 ± 8.9	4.7	165 ± 6.5	3.9	194 ± 14	7.0	182 ± 16	8.8
FA 22:5	Normal	73 ± 3.0	4.1	76 ± 1.8	2.4	82 ± 5.9	7.2	77 ± 5.3	6.8
	High	146 ± 8.5	5.8	123 ± 4.4	3.6	147 ± 9.9	6.7	139 ± 13	9.6
FA 22:6	Normal	197 ± 5.2	2.7	205 ± 5.3	2.6	216 ± 16	7.4	206 ± 12	6.0
	High	163 ± 8.5	5.2	141 ± 6.4	4.5	167 ± 14	8.6	157 ± 15	9.7

Values are average fatty acyl (FA) concentrations (µM) in human plasma from a fasted normoinsulinemic subject (representing normal total lipid level) and a 6 h postprandial hyperinsulinemic subject (representing high total lipid level). Plasma samples were hydrolyzed in the presence of H_2_^18^O to assess intra-day precision. Acid-catalyzed hydrolysis of normal and high lipid level samples was done on three consecutive days to assess inter-day precision. * one value out of five for “FA 18:2 Day 2 High insulin” has been removed after outlier correction. CV: coefficient of variation

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
