# Peer review of "Total Fatty Acid Analysis of Human Blood Samples in One Minute by High-Resolution Mass Spectrometry"

_biomolecules, 2018, doi:10.3390/biom9010007_

Round 1
Reviewer 1 Report
I will suggest to give more details in the title: "in one minute by high-resolution mass spectrometry"
Author Response
We are grateful to the reviewers for their thorough consideration and useful critique on our manuscript. We have addressed the reviewers’ concerns and revised the manuscript accordingly. A point-by-point response is provided below.
I will suggest to give more details in the title: "in one minute by high-resolution mass spectrometry"
We thank the reviewer for the excellent suggestion. Accordingly, we have changed the title.
Reviewer 2 Report
This is a very nice approach to high-throughput fatty acid analysis and will be a great addition to the lipidomics special edition.
The only point that requires some discussion is the consistent overestimation of 20:4 compared to the NIST reference concentration (Table S1).
Minor grammatical errors
Line 126: 210 μL of 155 mM
Line 128: with 7 μL of 162.3 μM PE 15:0/18:1(+[2]H7) added as internal standard
Line 129: 80 μL of freshly prepared
Line 130: For hydrolysis, the sample
Line 138 Ten μL of each fatty acid extract was infused
Line 139: an Orbitrap Fusion Tribrid mass spectrometer
Author Response
We are grateful to the reviewers for their thorough consideration and useful critique on our manuscript. We have addressed the reviewers’ concerns and revised the manuscript accordingly. A point-by-point response is provided below.
This is a very nice approach to high-throughput fatty acid analysis and will be a great addition to the lipidomics special edition.
The only point that requires some discussion is the consistent overestimation of 20:4 compared to the NIST reference concentration (Table S1).
In the revised manuscript we now comment on this discrepancy.
Minor grammatical errors
Line 126: 210 μL of 155 mM
This has been corrected.
Line 128: with 7 μL of 162.3 μM PE 15:0/18:1(+[2]H7) added as internal standard
This has been corrected.
Line 129: 80 μL of freshly prepared
This has been corrected.
Line 130: For hydrolysis, the sample
This has been corrected.
Line 138 Ten μL of each fatty acid extract was infused
This has been corrected.
Line 139: an Orbitrap Fusion Tribrid mass spectrometer
This has been corrected.
Reviewer 3 Report
Summary
Gallego et al. present a fast method to quantify fatty acids by direct infusion, high resolution shotgun lipidomics. By employing an acid catalyzed transesterification with 18O enrich water they could increase the signal to noise for palmitic and stearic acid which are common contaminants of organic solvents. Apart from determining the total fatty acid pool they additionally could follow the incorporation of 13C labeled palmitic acid in humans.
Broad comments
It is commonly known that most, even ultrapure organic solvents are contaminated with fatty acids (mostly C16:0 and C18:0) which limits their quantification. In the abstract the use of 18O enrich water is already discussed and in my opinion it should be mentioned why this approach was chosen.
Only a limited amount of plasma is needed for the analysis so is there a way to combine the fatty acid quantification with the usual shotgun lipidomics workflow?
Why was specifically PE 15:0/18:1 (+[2]H7) chosen?
What is the minimal resolution needed to differentiate the acid based transesterification products? Most labs will not have access to this high resolution.
The exact procedure of data analysis by ALEX should be explained in more detail. Are signals from single and double 18O labeled fatty acids added up?
It is mentioned that odd chain fatty acids are present in plasma, therefore it would be good to have a reference table of all detected and quantified fatty acids including odd chain ones.
Is the method really ready for clinical analysis in terms of reproducibility, accuracy, etc.? Would it be approved by FDA or any similar agency?
Specific comments
The nomenclature of deuterated lipid standards should be changed to something simpler. For instance PE 15:0-18:1 (d7). How was the position of fatty acids for this standard determined? If it was not determined please change it to the nomenclature avanti polar lipids is using.
What is the reason for the three microscans acquired?
Did you ever try ever to distill the solvents on your own to reduce the amount of free fatty acids?
Author Response
We are grateful to the reviewers for their thorough consideration and useful critique on our manuscript. We have addressed the reviewers’ concerns and revised the manuscript accordingly. A point-by-point response is provided below.
Summary
Gallego et al. present a fast method to quantify fatty acids by direct infusion, high resolution shotgun lipidomics. By employing an acid catalyzed transesterification with 18O enrich water they could increase the signal to noise for palmitic and stearic acid which are common contaminants of organic solvents. Apart from determining the total fatty acid pool they additionally could follow the incorporation of 13C labeled palmitic acid in humans.
Broad comments
It is commonly known that most, even ultrapure organic solvents are contaminated with fatty acids (mostly C16:0 and C18:0) which limits their quantification. In the abstract the use of 18O enrich water is already discussed and in my opinion it should be mentioned why this approach was chosen.
We thank the reviewer for the suggestion. In the revised manuscript we now state: ‘The resulting “mass-tagged” FA analytes can be specifically monitored with improved signal-to-background by 1 min of high resolution FTMS analysis on an Orbitrap-based mass spectrometer’.
Only a limited amount of plasma is needed for the analysis so is there a way to combine the fatty acid quantification with the usual shotgun lipidomics workflow?
Yes - in principle it would be possible to analyze a given plasma sample by both total FA analysis and also by another lipidomics workflow. However, we deem that it is beyond the scope of the manuscript to also present experimental data that exemplifies this possibility.
Why was specifically PE 15:0/18:1(+[2]H7) chosen?
No particular reason - it is a standard we also use for our shotgun lipidomic analyses of plasma and other tissues.
What is the minimal resolution needed to differentiate the acid based transesterification products? Most labs will not have access to this high resolution.
The cluster of isotopologues at m/z 309 in Figure 1F can be baseline resolved using a mass resolution of ~95,000 and above. In the revised manuscript we now state “ Notably, baseline separation of isobaric FA analytes can be achieved using instrumentations with a mass resolution higher than ~95,000 (FWHM at m/z 300) (data not shown).”
The exact procedure of data analysis by ALEX should be explained in more detail. Are signals from single and double 18O labeled fatty acids added up?
Yes - signals from FA analytes with one (16O18O) and two (18O2) O-18 atoms are added up. In the revised manuscript we now state ‘FA species were quantified by summing the intensities of doubly (18O2) and singly (16O18O) tagged FA analytes, normalizing to the sum intensity of doubly (18O2) and singly (16O18O) tagged internal standard FA 18:1(+[2]H7), and multiplying by the spike amount of FA 18:1(+[2]H7).’
It is mentioned that odd chain fatty acids are present in plasma, therefore it would be good to have a reference table of all detected and quantified fatty acids including odd chain ones.
In the revised manuscript we have now included a supplementary data file with all data pertaining to Figure 4 (SRM 1950), Figure 5 (SRM 2378) and Figure 7 (intake of NEFA 16:0(+13C16)).
Is the method really ready for clinical analysis in terms of reproducibility, accuracy, etc.? Would it be approved by FDA or any similar agency?
The performance characteristics (incl. reproducibility and accuracy) are as we have presented them the manuscript. In our opinion the method well-suited for clinical analysis. However, it is up to the individual investigator and institution to decide whether they want to use the method for clinical analyses and whether they want to further validate the method for approval by FDA or a similar agency.
Specific comments
The nomenclature of deuterated lipid standards should be changed to something simpler. For instance PE 15:0-18:1 (d7). How was the position of fatty acids for this standard determined? If it was not determined please change it to the nomenclature avanti polar lipids is using.
In the revised manuscript, in the Materials and Methods, we now specify the product name used by the vendor Avanti Polar Lipids (e.g., 1-pentadecanoyl-2-oleoyl(d7)-sn-glycero-3-phosphoethanolamine) alongside with the shorthand notation (e.g., PE 15:0/18:1(+2H7)). The position of fatty acyl chains is based on the product information provided by Avanti Polar Lipids. We note that isotopologues are now referenced as iAj - where A denotes the atom, i the total number of protons and neutrons and j denotes the number of atoms. As such, we do not use ‘d’ for deuterium but instead the more generic notation 2H.
What is the reason for the three microscans acquired?
To reduce size of acquired .RAW files.
Did you ever try ever to distill the solvents on your own to reduce the amount of free fatty acids?
No we did not. We did consider doing it, but deemed that this would only have a marginal effect as the FA contaminants are unfortunately omnipresent.